# Beyond six feet: The collective behavior of social distancing

Zhijun Wu [ORCID] *

Department of Mathematics, Iowa State University, Ames, Iowa, United States of America

* zhijun@iastate.edu

**Citation:** Wu Z (2024) Beyond six feet: The collective behavior of social distancing. PLoS ONE 19(9): e0293489. https://doi.org/10.1371/journal.pone.0293489

**Data Availability Statement:** All relevant data are within the manuscript and its Supporting information files.

**Funding:** This work is supported partially by the Simons Foundation Mathematics and Physical

## Abstract

In a severe epidemic such as the COVID-19 pandemic, social distancing can be a vital tool to stop the spread of the disease and save lives. However, social distancing may induce profound negative social or economic impacts as well. How to optimize social distancing is a serious social, political, as well as public health issue yet to be resolved. This work investigates social distancing with a focus on how every individual reacts to an epidemic, what role he/she plays in social distancing, and how every individual's decision contributes to the action of the population and vice versa. Social distancing is thus modeled as a population game, where every individual makes decision on how to participate in a set of social activities, some with higher frequencies while others lower or completely avoided, to minimize his/her social contacts with least possible social or economic costs. An optimal distancing strategy is then obtained when the game reaches an equilibrium. The game is simulated with various realistic restraints including (i) when the population is distributed over a social network, and the decision of each individual is made through the interactions with his/her social neighbors; (ii) when the individuals in different social groups such as children vs. adults or the vaccinated vs. unprotected have different distancing preferences; (iii) when leadership plays a role in decision making, with a certain number of leaders making decisions while the rest of the population just follow. The simulation results show how the distancing game is played out in each of these scenarios, reveal the conflicting yet cooperative nature of social distancing, and shed lights on a self-organizing, bottom-up perspective of distancing practices.

## Introduction

Social distancing has been advocated as an effective non-pharmaceutical measure to prevent the spread of epidemics [1–4], and has become especially well aware of to the public since the outbreak of the COVID-19 pandemic [5–13]. Social distancing plays a critical role to flat the epidemic curve before a vaccine becomes available, and remains to be so even after the population achieves certain herd immunity, for the pathogen always evolves, and the epidemic may develop into an endemic [14–22].

In general, social distancing or more rigorously speaking, physical distancing is referred to as for people to simply keep certain distances away from each other and avoid close in-person

Sciences Collaboration Grants for Mathematicians (Award Number: 586065). There is no additional external funding received for this study.

**Competing interests:** The authors have declared that no competing interests exist.

interactions [23–26]. However, in practice, it is not as easy to carry out as it is said. For in modern societies, social interaction is an indispensable part of human life. It is only natural for people to participate in various social or economic activities instead of avoiding them. Then, although important for fighting epidemics, social distancing comes with a price: It may cause social disruptions, economic losses, psychological stresses, etc. [27–37].

To carry out social distancing, one needs to manage and balance among multiple social activities—to keep some of them but reduce or completely close some others, in order to prevent the epidemic from further spreading while avoiding possible social, economic, or psychological consequences [38–50]. However, it is unclear how to optimize the activities though, often causing confusions and frustrations and resulting in overly reacted or failed distancing practices [51–57].

Different social activities may have different contact rates and hence different infection rates: Some have low contact rates but are socially isolating such as hiking, gardening, staying home, or reading. Some others are socially more involved but have close social contacts such as large gatherings, going to night clubs, going to shopping malls, or watching sports. In between, there are social activities that are essential to our daily life such as grocery shopping, cafeteria dinning, visiting friends, taking buses, or going to schools or workplaces. Given a set of social activities, an individual needs to make decision for how to participate in each of them, some probably with higher frequencies while others lower or even completely avoided, so that his/her close social contacts can be minimized at a least possible social or economic cost.

The decision of each individual may depend on or be influenced by the actions of all other individuals in the population. For example, if everybody decides to stay home, an individual may choose to go out and have dinner at a restaurant although the risk of having close contacts at a restaurant usually is high. On the other hand, if the whole population decides to go hiking, he/she may want to avoid it although hiking is usually a low contact activity. Collectively, social distancing can thus be considered as a population game, where based on what the population does, every individual makes his/her own decision on how to participate in a given set of social activities so that he/she can minimize his/her social contacts and possible social or economic costs. An optimal distancing strategy can then be obtained when the game reaches an equilibrium [58, 59].

Research on social distancing has surged since the outbreak of the COVID-19 pandemic, providing a wealth of knowledge and experience on non-pharmaceutical measures for preventing epidemics from spreading. However, most of these studies are about the influences of public policies, economic concerns, or cultural differences on social distancing, but not particularly about social distancing as a human behavior or the collective behavior of a population for that matter, as pointed out in a recent article by Vardavas et al. 2021 [60]. In fact, human responses to and influences on public health measures such as mask wearing, testing, and vaccination have long been investigated in behavioral sciences based on the theory of planned behavior [61, 62] and the health belief models [63, 64], with great insights into how health behaviors may be perceived and carried out at both individual and population levels.

Work on mathematical modeling of social distancing has been pursued in the past [65–68] including some done recently [69–82], but the focus is on the dynamics of epidemics with changing patterns or levels of social distancing, with little specifics on how the distancing activities are carried out and how certain distancing patterns or levels are achieved. The work in this paper follows a game theoretic approach to social behavior in general [83–88] and to social distancing in particular [89–93], and investigates social distancing with a focus on how every individual reacts to an epidemic, what role he/she plays in social distancing, and how the individual decision contributes to the action of the population and vice versa. The collective behavior of social distancing is modeled as a population game, where every individual makes a

distancing decision and together the population reaches an equilibrium when every individual achieves his/her distancing goal.

A number of issues rise immediately for the general game model for social distancing: First, for a general population game, it is assumed that every individual interacts with all others in the population and knows their strategies, which is not true in the real world, where people usually interact only with their social acquaintances [94, 95]. It is also assumed that the individuals are all the same when evaluating the distancing risks and making their distancing decisions, but in reality, they are not. For example, children and adults seem to have different infection rates for COVID-19 and would therefore perceive the distancing risks of social activities differently [96, 97]; and so would the vaccinated and unprotected individuals, and the economically secure and vulnerable [98, 99]. In a general game, every individual is also required to be able to make rational decisions for the game to eventually reach equilibrium. The condition is again unrealistic, for not everyone is able to or willing to make his/her own decisions [100–102].

However, all these issues can be addressed by making several refinements on the general model: First, the population can be assumed to be distributed over a social network, and the decision of each individual can be made through the interactions with his/her social neighbors [103]. Such a network can be simulated by generating a small-world network using for example the Watts-Strogatz algorithm [104]. The simulation results presented in this paper show that the distancing game can be played successfully on such a network. Surprisingly, the game approaches to an equilibrium state as in the general case even when the interactions among the individuals are restricted only to their close neighbors.

Second, the population can be divided into different social groups according to certain social/biological/medical characteristics such as the age of the individuals (e.g., children, adults, seniors, etc.) or the level of protection (e.g., the vaccinated, recovered, unprotected, etc.) or the economic vulnerability (income above average, middle income, low income, etc.). The distancing risks of participating certain social activities can then be evaluated using different criteria for different social groups. Theoretical and simulation results for the distancing game in such heterogeneous populations are discussed in the paper, showing that the game can be played in almost the same form as in the general case: just use different risk-assessment functions for different social groups; and if the interactions among the individuals are more frequent inside than across social groups, each social group would eventually find its own equilibrium strategy while the whole population approaches to the average one.

Third, a certain number of individuals can be selected to act as leaders and the rest of the population as followers. The leaders make decisions on their own strategies while the followers simply copy the strategies of the leaders. The distancing game can then be carried out with such a leader-follower scheme. The simulation results show that the game can indeed proceed without requiring every individual to make rational decisions, and reach its equilibrium successfully when only a certain number of individuals, say 30% of the population, are designated as leaders. Indeed, in practice, it is likely that a certain number of individuals such as public health experts or community leaders make some decisions or recommendations while others follow.

As such, the work in this paper confirms the game model as a plausible approach to the study of the collective behavior of social distancing. The work is still at a theoretical development stage with the model yet to be further refined against real data. It nonetheless offers some insights into how social distancing, as an adaptive social behavior, is carried out at both individual and population levels, in complex social networks, and in heterogeneous populations. It reveals the conflicting yet cooperative nature of social distancing, and sheds lights on a self-organizing, bottom-up perspective to social distancing practices.

## Results

### As a population game

Consider a population of $m$ individuals with $n$ social activities. Assume that every individual needs to decide a frequency to participate in each of the activities, say $x_i$ for activity $i$. Then the collection of these frequencies $x = \{x_i: i = 1, \ldots, n\}$ can be considered as a distancing strategy of the individual. Let $y_i$ be the average frequency of the population to participate in activity $i$. Then, the collection of these average frequencies $y = \{y_i: i = 1, \ldots, n\}$ can be considered as a distancing strategy of the population. Here a frequency $x_i$ or $y_i$ can be represented by the active hours in activity $i$ in a week (total 112 hours per week if 16 hours are counted as active hours per day excluding 8 hours sleeping time).

Given a distancing strategy $y$ of the population, assume that the potential distancing risk of having close social contacts and negative social or economic impacts in activity $i$, when fully participated, can be represented by a function $p_i(y)$. Then, the distancing risk of an individual of strategy $x$ at activity $i$ must be $x_i p_i(y)$, and at all the activities together be $\Sigma_i x_i p_i(y)$, where $\Sigma_i$ means the sum over all $i$'s. Let this summation be denoted as a function $\pi(x, y)$. A distancing game can then be defined for an individual against the population with $\pi(x, y)$ as the cost function; and a strategy $x^*$ is an equilibrium strategy for the game if and only if every individual in the population takes this strategy (and hence $y^* = x^*$), and his/her distancing risk $\pi(x^*, y^*)$ using strategy $x^*$ is no greater than the distancing risk $\pi(x, y^*)$ using any other strategy $x$.

Assume that the social activities are independent of each other, i.e., the individuals participating in one activity do not have contacts with those in other activities. Then, the function $p_i(y)$ can be defined to depend only on the participating frequency $y_i$ of the population in activity $i$. Let $p_i(y) = w_i \sigma_i(y_i)$, where $w_i$ is a constant called the risk factor of activity $i$, and $\sigma_i$ is a logistic function of $y_i$, typically increasing slowly when $y_i$ is in a low range, picking up the speed after $y_i$ passes a certain threshold, and slowing down again when $y_i$ enters in a high range (as shown in Fig 1), which presumably corresponds to how the potential distancing risk at a given activity increases with increasing participating frequency of the population in that activity.

Now, consider a population state, called a complete social distancing state, when every individual minimizes his/her total distancing risk over all the activities with an equilibrium strategy $x^*$. Given $y^* = x^*$, a corresponding set of values for $w_i$ can be determined retrospectively (details in **Deriving contact and impact factors** in Methods). Set $\alpha_i$ to this $w_i$ for all $i$. Then, a set of parameters $\alpha_i$ is obtained, with $\alpha_i$ named as the contact factor of activity $i$, for in a complete social distancing state, the risk of having close social contacts is presumably minimized the most by using this set of parameters.

Then, consider another population state, called a free of social distancing state, when every individual also minimizes his/her total distancing risk over all the activities with another equilibrium strategy $x^*$. Given $y^* = x^*$, a corresponding set of values for $w_i$ can be determined again retrospectively (details in **Deriving contact and impact factors** in Methods). Set $\beta_i$ to this $w_i$ for all $i$. Then, another set of parameters $\beta_i$ is obtained, with $\beta_i$ named as the impact factor of activity $i$, for in a free of social distancing state, the risk of having negative social or economic impacts from social distancing is presumably minimized the most by using this set of parameters.

Table 1 shows the values of $\alpha_i$ and $\beta_i$ estimated for a small set of so-called commonly attended social activities or CASA activities for short. These activities are assumed to be typical in daily small town lives in North America and are grouped into 20 general categories for the testing and simulation purposes in this work. In practice, they can certainly be extended to a much larger set of more specific and refined activities.

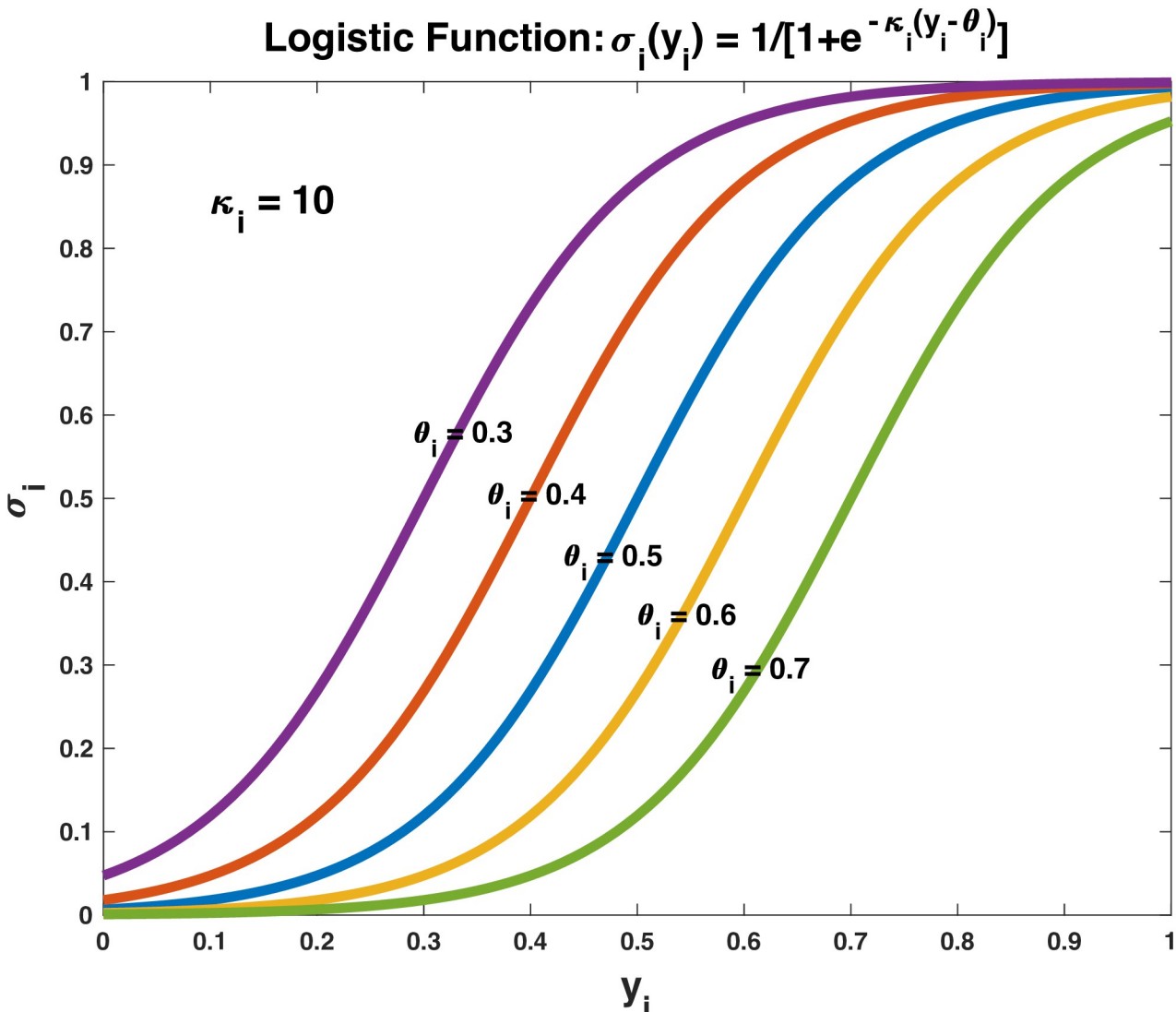

**Fig 1. Logistic functions** $\sigma_i(y_i)$ **with a rate constant** $\kappa_i = 10$ **shifted by** $\theta_i = 0.3, 0.4, 0.5, 0.6, 0.7$, **all increasing slowly when** $y_i$ **is in a low range, picking up the speed after** $y_i$ **passes a certain threshold, and slowing down again when** $y_i$ **enters in a high range.**

**Table 1. Estimated contact factors and impact factors for 20 CASA activities.**

| Act: | 1 | 2 | 3 | 4 | 5 | 6 | 7 | 8 | 9 | 10 |
|---|---|---|---|---|---|---|---|---|---|---|
| $\alpha_i$: | 1.0000 | 1.0000 | 1.0000 | 1.0000 | 1.8483 | 1.8483 | 1.8483 | 1.8483 | 2.4090 | 2.4090 |
| $\beta_i$: | 2.2052 | 2.2052 | 2.6318 | 2.6318 | 2.2052 | 2.6318 | 2.6318 | 2.2052 | 2.2052 | 2.6318 |
| **Act:** | **11** | **12** | **13** | **14** | **15** | **16** | **17** | **18** | **19** | **20** |
| $\alpha_i$: | 2.4090 | 2.4090 | 2.8754 | 2.8754 | 2.8754 | 2.8754 | 3.1418 | 3.1418 | 3.1418 | 3.1418 |
| $\beta_i$: | 2.6318 | 2.2052 | 2.4090 | 2.4090 | 0.8402 | 0.5948 | 2.0188 | 2.0188 | 2.4090 | 2.4090 |

Act—activity: 1—reading or watching TV, 2—work at home, 3—hiking, 4—gardening, 5—stay with family, 6—grocery shopping, 7—go to hospitals, 8—visit friends, 9—restaurant/cafeteria dinning, 10—go to shopping malls, 11—take buses, 12—go to churches, 13—watch sports, 14—attend concerts, 15—go to schools, 16—go to workplaces, 17—large gathering, 18—go to bars or night clubs, 19—air traveling, 20—go to movie theaters; $\alpha_i$—contact factors; $\beta_i$—impact factors.

The first eight CASA activities include 1. reading/watching TV, 2. work at home, 3. hiking, 4. gardening, 5. stay with family, 6. grocery shopping, 7. go to hospitals, 8. visit friends. They have low contact rates and hence relatively low contact factors. The last eight CASA activities include 13. watch sports, 14. attend concerts, 15. go to schools, 16. go to workplaces, 17. large gathering, 18. go to bars or night clubs, 19. air traveling, 20. go to movie theaters. They have high contact rates and hence relatively high contact factors. In between, there are also four frequently participated CASA activities, 9. restaurant/cafeteria dinning, 10. go to shopping malls, 11. take buses, 12. go to churches. Their values of contact factors are at a moderate level.

The values for the impact factors vary in a small range except for activities 15. go to schools and 16. go to workplaces whose impact factors are quite small. The smaller the impact factor, the less negative social or economic impact in the activity or in other words, the more socially or economically favorable. The values of the impact factors are not exactly negatively correlated with those of the contact factors. For example, although in the same level of contact factors, activities 3. hiking and 4. gardening have relatively larger impact factors than activities 1. reading/watching TV and 2. work at home which seem to be more socially or economically favorable. On the other hand, although activities 19. air traveling and 20. go to movie theaters have the largest contact factors, their impact factors are not the smallest. Instead, the impact factors for activities 15. go to schools and 16. go to workplaces are among the smallest, agreeing with the fact that they are two of the most important social and economic activities in modern human lives.

Now, let $w_i = \delta_i \alpha_i + (1 - \delta_i)\beta_i$, $0 \le \delta_i \le 1$. A general set of risk factors can be obtained with $\delta_i$ called a severity parameter. If $\delta_i = 1$ for all $i$, $w_i = \alpha_i$. If $\delta_i = 0$ for all $i$, $w_i = \beta_i$. If $0 < \delta_i < 1$, $w_i$ determines an equilibrium strategy $x^*$ and hence $y^*$ that corresponds to a population state between complete and free of social distancing. Based on general theory for population games, given $p_i(y) = w_i \sigma_i(y_i)$, if $w_i > 0$ for all $i$, an equilibrium strategy $x^*$ for the distancing game can be obtained with $x_i^* = \sigma_i^{-1}(\lambda/w_i)$, $i = 1, \ldots, n$, assuming $x_i^* > 0$ for all $i$, where $\lambda$ is a constant such that $\Sigma_i \, \sigma_i^{-1}(\lambda/w_i) = 1$ (details in **Equilibrium strategies and stabilities** in Methods).

For convenience, let the same logistic function $\sigma_i$ with $\kappa_i = 10$ and $\theta_i = 0.5$ be used for all the activities in Table 1. Then, the equilibrium strategies of the distancing game for this set of activities can immediately be computed for some fixed value of $\delta_i$. Fig 2 illustrates the contrast among three of these equilibrium strategies with $\delta_i = 1$, 0, or 0.5 for all $i$, corresponding to the games for complete social distancing, no social distancing, or partially social distancing, respectively. For complete social distancing (blue circles, $\delta_i = 1$) when close social contacts are supposed to be reduced the most, for the first eight CASA activities whose contact rates are low, the participating frequencies are high, while for the last eight activities whose contact rates are high, the participating frequencies are low. On the other hand, if free of social distancing (red circles, $\delta_i = 0$) when negative social or economic impacts of social distancing are contained the most, for the first eight activities, the participating frequencies are much lower than those for complete social distancing, while for the last eight activities, the participating frequencies are much higher. In between when both contacts and impacts are concerned (brown plus signs, $\delta_i = 0.5$), for the first eight activities, the participating frequencies are still higher than those free of social distancing, but not as high as those for complete social distancing, and for the last eight activities, the participating frequencies are certainly lower than those free of social distancing, but not as low as those for complete social distancing.

## In small-world social networks

To be more realistic, assume that the population is distributed over a social network, and each individual only interacts with his/her neighbors in the social network. The distancing game

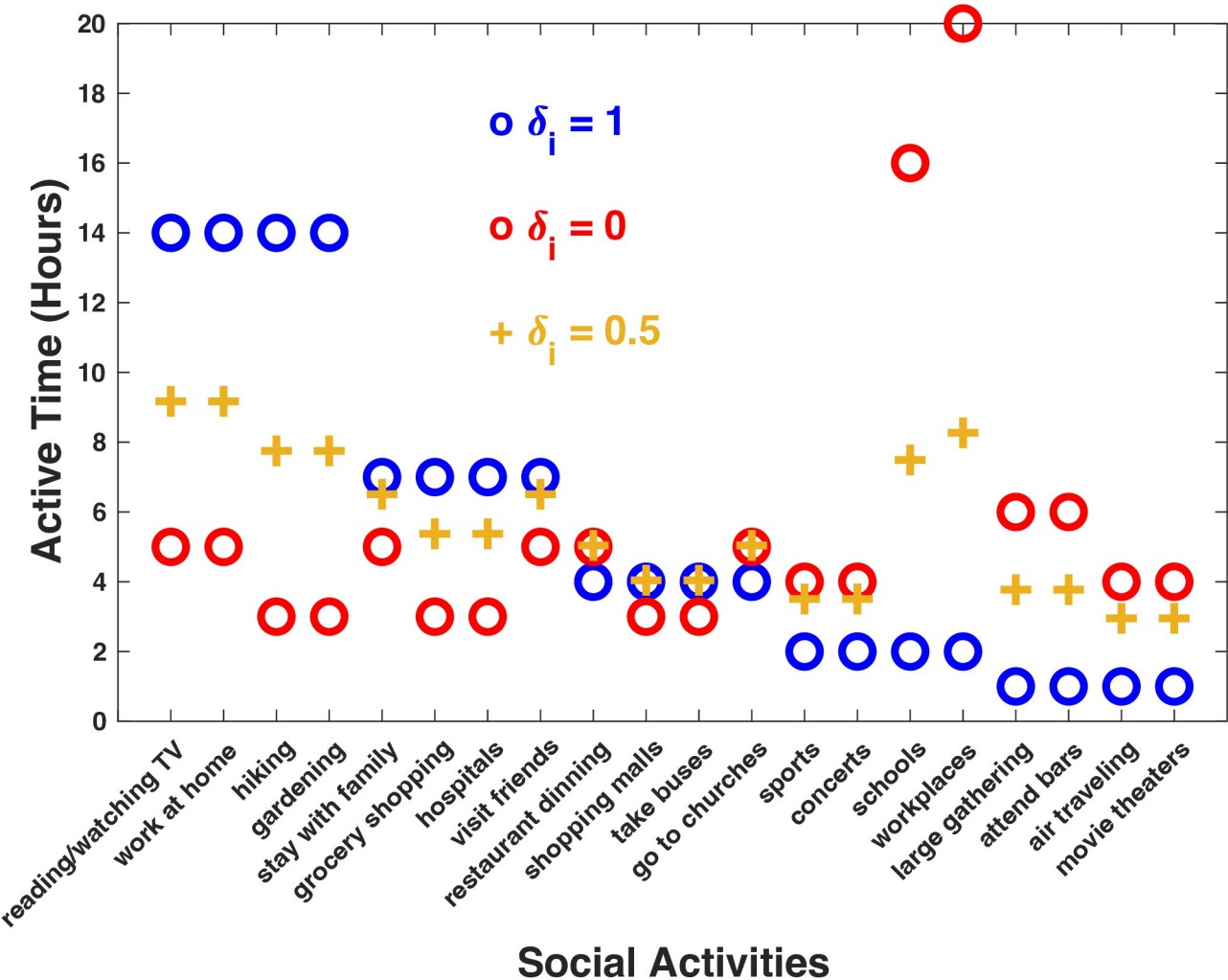

**Fig 2. The equilibrium strategies of the distancing game for complete social distancing (blue circles), free of social distancing (red circles), and partially social distancing (brown plus signs), displayed as active times in hours per week (total 112 active hours).**

can then be viewed as one played by every individual against the population in his/her neighborhood in the social network. Depending on the neighborhood size, the game becomes against a fraction of population ranging from the immediate neighbors of the individual to the whole population. For convenience, define the neighborhood of size $k$ of an individual to be one that includes all the neighbors connected to the individual with up to $k$ consecutive links. Then, if $k$ is large enough, the neighborhood would include the whole population.

To mimic a real social network, the Watts-Strogatz algorithm [104] is used to generate a small-world social network. Assume that there are 200 individuals in the population ($m = 200$), the average degree of the nodes is 6 ($K = 6$), and 30 percent of all the links for each node come from random connections ($b = 0.3$). Fig 3 shows the generated network and the distribution of the degrees of the nodes. The nodes are displayed around a circle. About 70 percent of the links are along the edges which are not clearly visible in the graph. The rest of the

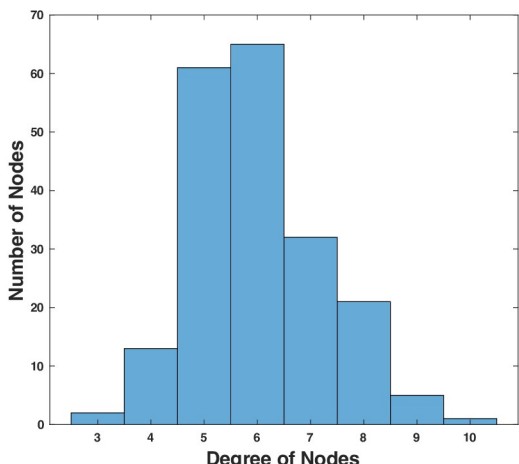

**Fig 3. A small-world social network generated by using the Watts-Strogatz algorithm.**

links are more visible in the interior of the circle. In general, as the degree parameter $K$ or the randomness parameter $b$ increases, the graph becomes denser with more interior links (details in **Generating small-world social networks** in Methods).

Assume that every individual plays the distancing game with the population in a neighborhood of the same size. Let $x$ be the strategy of any individual and $y$ the average strategy in the corresponding neighborhood. Then, the potential distancing risk at activity $i$ can still be estimated using the function $p_i(y)$, with which the individual can update his/her own strategy $x$: If $p_i(y)$ is higher than the average over all other activities, activity $i$ must be riskier, and $x_i$ in activity $i$ should be decreased; otherwise be increased. The update can be repeated for every individual in the population until no one can further improve his/her strategy, and the game hopefully reaches an equilibrium.

The distancing game is simulated on networks similar to the one in Fig 3. The simulation algorithm is outlined in **Simulation of distancing games in social Networks** in Methods. The parameters for the networks are fixed to $m = 2000$ and $K = 6$. The activities in Table 1 are used for the game and therefore, $n = 20$. The same logistic function $\sigma_i(y_i)$ with $\kappa_i = 10$ and $\theta_i = 0.5$ is used in $p_i(y) = w_i\sigma_i(y_i)$ for all $i$. Other parameters for the game are varied with the severity parameter $\delta_i = 0, 0.25, 0.5, 0.75, 1$ for all $i$, the randomness parameter for the network $b = 0.10$, 0.20, 0.30, 0.40, 0.50, and the neighborhood size $k = 1, 2, 3, 4, 5, 6$. For each set of parameters, an equilibrium strategy $x^*$ for every individual and $y^*$ for the population for the general distancing game can be computed directly as described in the previous section. Let $x$ be the strategy for an individual obtained by the simulation and $y$ the corresponding neighborhood strategy. These quantities are recorded in the simulation and compared with the equilibrium strategies $x^*$ and $y^*$. The results for all the parameter settings are documented in S1 Text.

The simulation runs in multiple iterations, each named as a generation. In each generation, every individual gets a chance to update his/her strategy followed by an adjustment on the population strategy. The simulation ends when every individual strategy $x$ becomes very close to the equilibrium strategy $x^*$ of the game. The closeness is measured by a so-called Euclidean distance $\|x - x^*\|$ between the two strategies, $\|x - x^*\| = \sqrt{\Sigma_i (x_i - x_i^*)^2}$. The simulation is considered to be converged if the average Euclidean distance $<\|x - x^*\|>$ between $x$ and $x^*$ in the whole population is smaller than a prescribed small number, say $10^{-4}$.

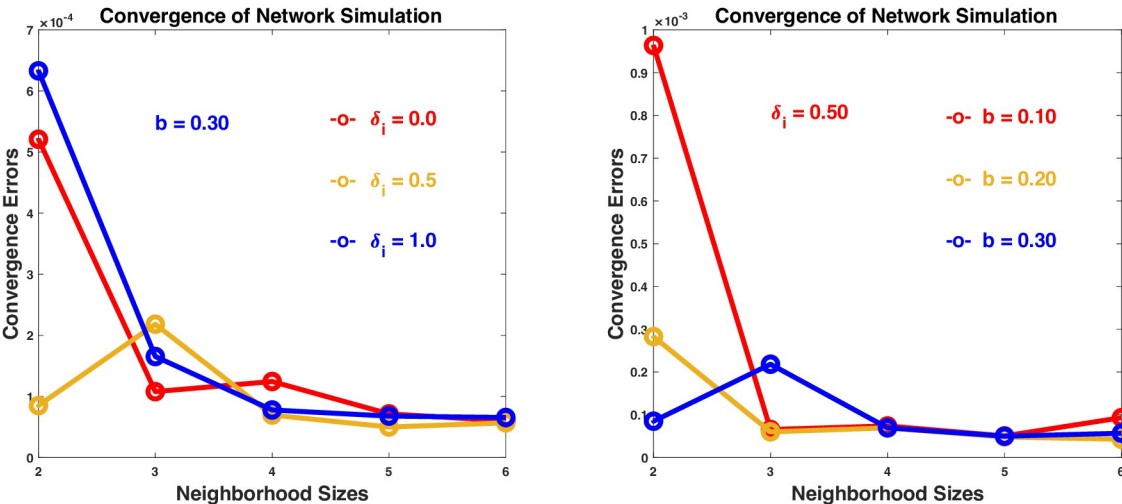

**Fig 4. Convergence of individual strategy $x$ obtained from network simulation to the equilibrium strategy $x^*$ of the distancing game.** The convergence error is measured by the average Euclidean distance $<\|x - x^*\|>$ between $x$ and $x^*$ in the population, where $\|x - x^*\|^2 = (x_1 - x_1^*)^2 + \cdots + (x_n - x_n^*)^2$, and $<>$ means the average over all the individuals.

Throughout the simulation, the game is tested for different neighborhood sizes $k = 1, 2, 3,$ 4, 5, 6 with all other parameters being fixed. The game converges for all the neighborhood sizes of $k = 2, 3, 4, 5, 6$ except for $k = 1$ when the number of neighbors seems to be too small for an individual to interact with and make decision upon. This is true for all different randomness values $b = 0.10, 0.20, 0.30, 0.40, 0.50$ for the network and severity parameters $\delta_i = 0.0,$ 0.25, 0.50, 0.75, 1.00 for risk assessment. The game is started with every individual assigned to an initial strategy $x$ randomly generated around a strategy randomly perturbed from $x^*$. It is tested for the initial strategies generated with 0%, 10%, 20%, 30%, 40%, 50% perturbations of $x^*$. The game converges for all the initial strategies with $\leq 40\%$ perturbations, most requiring only less than 10 generations (details in S1 Text).

Fig 4 demonstrates the network simulation results for two typical scenarios: In Fig 4(a), the convergence results are shown for the game with $\delta_i = 0.0, 0.5, 1.0$ and $k = 2, 3, 4, 5, 6$ while $b$ is fixed to 0.30. In Fig 4(b), the results are shown for the game with $b = 0.10, 0.20, 0.30$ and $k = 2,$ 3, 4, 5, 6 while $\delta_i$ is fixed to 0.5. In both cases, for different values of $\delta_i$ and $b$, the game converges for all different neighborhood sizes greater than or equal to 2. The accuracy for $k = 2$ is not very high but acceptable. It improves as $k$ becomes larger. When $b$ is fixed to 0.30 but $k$ is changed from 1 to 2, 3, 4, 5, 6, the average number of neighbors for each individual changes from 7 to 29, 116, 425, 1187, 1897. The last number agrees with the general consensus on small-world social networks where almost every pairs of individuals can be found connected with up to more or less 6 consecutive links [105–107]. It is therefore not surprising that when $k = 6$, the game on this network converges to the same equilibrium strategy as the general distancing game, for it is almost the same as the general distancing game played by every individual against the whole population. It is surprising, however, that when the neighborhood size is reduced, even when $k = 2$ with only 29 neighbors in average for each individual, the game still converges to the equilibrium strategy of the general distancing game, with $x$ converging to $x^*$.

## In heterogeneous populations

Not every individual is equally vulnerable for epidemic infection. Nor is every individual equally likely to spread the disease. For example, for COVID-19, children seem not as

susceptible to infection as adults, and they are less likely to carry and spread the virus [96, 97]. Similarly, the vaccinated people are more or less immune to infection and are probably more free to participate in social activities than those unvaccinated. Furthermore, the essential workers or economically vulnerable individuals are more concerned with the social or economic impacts than the health benefits of social distancing [98, 99]. A population should therefore be divided into different groups who perceive the distancing risks of social activities differently.

As an example, consider a population evenly divided into 4 population groups: either according to the ages of the individuals into $g_1$: 1–20 years old; $g_2$: 21–40; $g_3$: 41–60; and $g_4$: 61–80, or according to the level of protection of the individuals into $g_1$: the vaccinated; $g_2$: the recovered; $g_3$: the unprotected; and $g_4$: the most vulnerable. Set $\delta_i$ for each of the groups to 0.00 for $g_1$, 0.25 for $g_2$, 0.75 for $g_3$, and 1.00 for $g_4$, thereby making the first two groups more open to expand their social activities but the last two to prefer more social distancing. Fig 5 demonstrates the results from computer simulation for the distancing game played among these population groups. The simulation is conducted in a similar setting as for the game on the network shown in Fig 3 with $m$ = 2000, $K$ = 6, $b$ = 0.30, and $k$ = 3. In each generation of the simulation, every individual plays the game once, i.e., has a chance to update his/her strategy. However, different from the simulation described in the previous section, when evaluating the potential distancing risks, the contributions from different population groups are different due to their different $\delta_i$ values. When they are counted, more weight is also given to the contribution from the individual's own group than from other groups. In addition, when comparing with the population average on the distancing risks, only the average over the individual's own group is considered (details in **Distancing in heterogeneous populations** in Methods).

Fig 5 displays four snapshots from the simulation, showing the changes of the individual strategies of the game in four different generations. The circles represent the individual strategies and the stars the average population strategies. The circles are color coded for different population groups, with red for $g_1$, magenta for $g_2$, cyan for $g_3$, and blue for $g_4$. For each activity, there are 2000 circles corresponding to the participating frequencies of 2000 individuals for the activity. The labels on the $x$-axis are 20 CASA activities as defined in Table 1. The first graph in the figure shows the strategies of the individuals at the beginning of the 1st generation of the simulation, which are randomly generated around reasonably guessed starting strategies for each of the population groups. The second graph shows the strategies of the individuals after the 3rd generation, when they start separating into different groups. The third graph shows the strategies after the 6th generation, when they almost converge to their equilibrium positions. The last graph shows the strategies after the 9th generation, when they are close enough to their equilibrium values, and the simulation is terminated.

In the end of the simulation, each population group reaches an equilibrium strategy or more rigorously, its approximation. Since $\delta_i$ = 0.00 and 0.25 for all $i$ for $g_1$ and $g_2$, the individuals in these two groups are considered to be more risk-taking, and their frequencies to participate in the socially active though high-contact activities (13–20) appear to be higher than the population average, while their frequencies to stay with the low-contact but socially isolating activities (1–8) are lower than the population average. On the other hand, since $\delta_i$ = 0.75 and 1.00 for all $i$ for $g_3$ and $g_4$, the individuals in these two groups are considered to be more conservative, and their frequencies to stay with the low-contact though socially isolating activities (1–8) appear to be higher than the population average, while their frequencies to join the socially active but high-contact activities (13–20) are lower than the population average.

Simulations for distancing games with different population groups are conducted with varying severity parameter $\delta_i$, randomness parameter $b$, and neighborhood size $k$. The results from these simulations (documented in S2 Text) are all consistent with what are observed in the above example, showing that the game model can be extended to heterogeneous

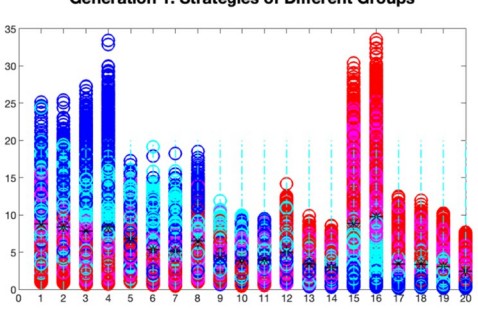

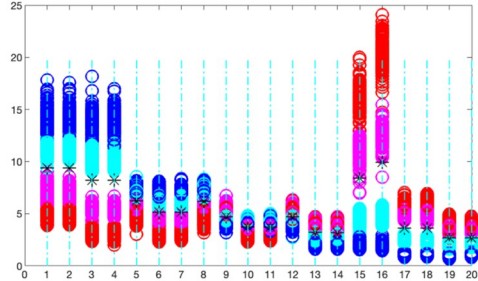

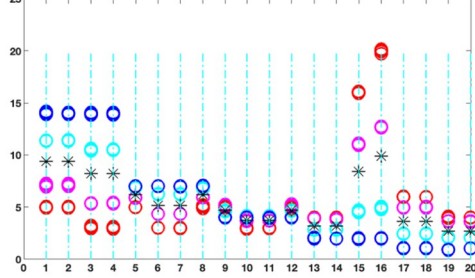

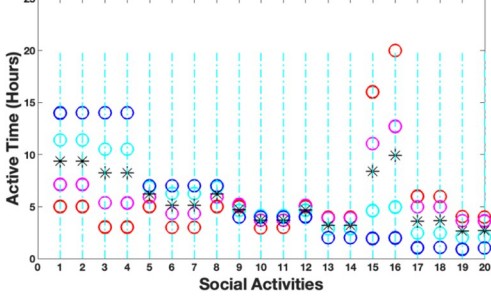

**Fig 5. The simulation results for an example distancing game with a heterogeneous population.** Shown in the figure are the individual and population strategies of the game in four different generations of the simulation. The stars represent the population strategies and the circles the individual strategies. The circles are color coded for different population groups, with red for $g_1$, magenta for $g_2$, cyan for $g_3$, and blue for $g_4$. Over each activity, there are 2000 circles corresponding to the strategies for 2000 individuals. Labeled in the x-axis are 20 CASA activities. Along the y-axis are the active times in hours per week.

populations to predict different distancing behaviors among different population groups. What unexpected in these results is that the game for each population group converges to its own equilibrium strategy as if it is played by each group alone, while the population strategy is simply a collective result of all these individual group strategies (details in **Distancing in heterogeneous populations** in Methods). This may not be surprising, on the other hand, as observed during the COVID-19 pandemic that children and adults do have their own distancing strategies as their shares in the average distancing strategy of the population.

## By following the leaders

Not every individual actively participates in social distancing. Even if he/she does, he/she may not necessarily make the decisions as accurately as assumed such as evaluating the distancing risks of the activities and responding with appropriate actions, etc. In practice, it is likely that a certain number of individuals such as public health experts or community leaders make some decisions or recommendations while others follow [100–102]. Indeed, leadership plays an important role in collective actions in both nature and human societies [108–112].

In order to incorporate the leadership factor into the distancing model, a certain number of individuals are designated randomly as leaders and the rest of the population as followers. A leader makes a distancing decision as a regular player in the distancing game, while a follower just copies the strategies of some leaders unless he/she cannot find a leader in his/her group among his/her closest neighbors when he/she either makes his/her own decision or simply follows the crowd (details in **Following the leaders vs. following the crowd** in Methods).

The game with mixed leaders and followers is simulated in a small-world social network similar to the one in Fig 3 with varying neighborhood sizes and percentages of leaders in the population. It is also assumed to be against a heterogeneous population as given in the example game in the previous section, where there are four population groups, and the first two groups contribute to the distancing risks differently from the last two. Fig 6 shows some simulation results with $m = 2000$, $K = 6$, $b = 0.30$, and $k = 3$ for the network but varying percentages of leaders in the population.

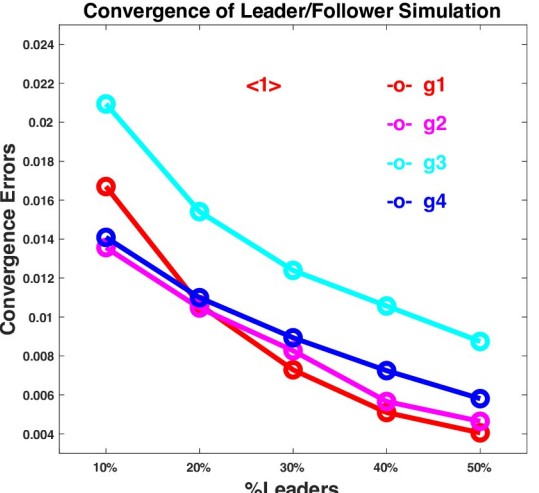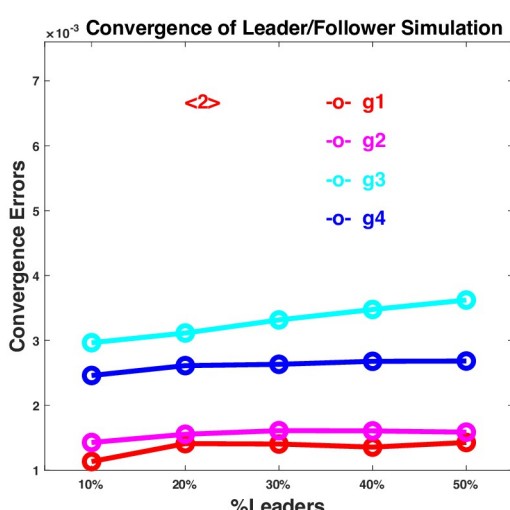

**Fig 6. Convergence of individual strategy $x$ obtained from leader/follower simulation to the equilibrium strategy $x^*$ of the distancing game.** The convergence error is measured by the average Euclidean distance $<\|x - x^*\|>$ between $x$ and $x^*$, where $\|x - x^*\|^2 = (x_1 - x_1^*)^2 + \cdots + (x_n - x_n^*)^2$, and $<>$ means the average over all the individuals. <1>—results by following the crowd if leaders not found; <2>—results by self-determination if leaders not found.

The first set of results <1> in Fig 6(a) is obtained with a follow-the-crowd strategy if a follower cannot find a group leader among his/her closest neighbors. When there is a high percentage of leaders in the population ($\geq$ 30%), the game converges to the equilibrium strategies in reasonable accuracies for all population groups. If there are lower than 30% leaders in the population, the convergence becomes less accurate, when most of the followers are not able to find a group leader in their neighborhood, and there is a disadvantage by simply following the crowd.

The second set of results <2> in Fig 6(b) is obtained with a make-own-decision strategy if a follower cannot find a group leader among his/her closest neighbors. Similar to the previous case, when there is a high percentage of leaders in the population ($\geq$ 30%), the game converges to the equilibrium strategies in reasonable accuracies for all population groups. In contrast to the previous case, when the percentage of leaders are less than 30%, the convergence remains to be as accurate, because when there are fewer leaders, more followers start making their own decisions, which can be even better than following the leaders.

The simulation results with other neighborhood sizes and percentages of leaders (documented in S3 Text) are all consistent with those in the above example. In general, when the percentage of leaders in the population is high, the game is expected to perform well and reach its equilibrium strategy. When the percentage is not very high, the game still runs reasonably well, showing that a group of leaders are able to guide, and not every individual is required to make a decision. However, as shown in contrast between the results in <1> and <2>, when there are fewer leaders, making own decisions actively is certainly more reliable than simply following the crowd.

## Discussion

The social distancing activities are not easy to track and hence difficult to study experimentally. The model proposed in this work presents a theoretical framework with which the distancing behaviors can be simulated, predicted, and analyzed. In this model, every individual in a given population is assumed to engage in social distancing by playing a distancing game, where based on what everybody else does, every individual makes decision on how to participate in a given set of social activities so that he/she can minimize his/her close social contacts with least possible negative social or economic impacts.

The model is built by referring to two possible population states where the distancing risks of a set of social activities are assessed. First, a complete social distancing state is assumed, and the corresponding risk factors are extracted as the contact factors. Then, a free of social distancing state is assumed, and the corresponding risk factors are derived as the impact factors. The contact factors are correlated with but not equivalent to the contact rates. They are so named as in a complete social distancing state, the close social contacts are perceived as the main social risks to reduce. Similarly, the impact factors are not directly related to the social or economic costs of the activities. They are so named as in a free of social distancing state, the negative social or economic impacts are the main concerns to minimize. By combining the two sets of parameters, a general set of risk factors can then be defined for a population in between the two extremum states.

As for any general population game, for the general distancing game, it is assumed that every individual interacts with all others in the population as discussed in **Results—As a population game**. To be more realistic, in this work, the population is then assumed to be distributed over a social network, and each individual interacts only with his/her social neighbors. An algorithm is implemented to simulate the distancing game on such a network as discussed in **Results—In small-world social networks**. The algorithm is not equivalent to the general

distancing game, and is not guaranteed to converge to an equilibrium strategy, either. In addition, the distancing game on a social network is played by every individual against his/her neighborhood, not the whole population. Nonetheless, the games simulated on social networks all converge to the equilibrium strategies of the general distancing games even when the neighborhood sizes are small and are restricted to contain only close neighbors, which is surprising, or not, as it may be how it is played out in the real world.

The extension of the distancing game to populations with multiple population groups in **Results—In heterogeneous populations** allows each population group to have its own assessment on distancing risks and hence its own distancing strategy. In such a game, the individuals take into account the contributions to their distancing risks from all population groups while giving more weights to those from their own groups, as they are supposed to interact more with the individuals in their own groups than those in other groups. In theory, the game converges to an equilibrium strategy, with each group having a strategy as if it plays the game alone as justified in **Appendix—Games with multiple population groups**. This property is further observed in the simulation when the population is spread on a social network. It reveals a critical condition under which multiple social groups compete yet maintain their own independent strategies. It applies well to social distancing activities in heterogeneous populations. Future work along this line may be extended to the effects on social distancing from the homophilic structure of the population and the clustering structure of the social network as studied in recent work in [113–115].

The leadership role in social distancing is addressed lightly in this work as discussed in **Results—By following the leaders**. In fact, the leadership in social distancing is way beyond a matter of a number of leaders making their own distancing decisions. For better or worse, leadership is often a determining factor in directing or changing the social distancing activities, as local or global organizations or governments make public health policies, provide social distancing guidelines, or give lockdown orders, etc., which are out of the scope of this study. However, the work in this study may help to understand the nature of social distancing as a collective behavior of human population, thereby providing a quantitative approach to assessing and improving the outcomes of public health policies concerning the control of social activities and its potential impacts.

Social behaviors including health-related behaviors have been studied for long time in behavioral sciences, most notably, in the theory of planned behavior [61, 62] and the health belief models [63, 64]. The work in this paper is not directly based on these theories but is consistent with their general principles. In the health belief models, the individual responses to health concerns are considered to be based on the outcomes of the assessments on the perceived susceptibility to the ill, perceived severity of the ill, perceived barriers to take actions, and perceived benefits of behavioral changes [63, 64]. The estimates on the distancing risks in the game model can be considered as the results from similar assessments, although not explicitly correlated to them. Since the game model is focused more on the strategies of participating in social activities, the risk assessments depend more on the selected activities as well as the distribution of the population over the activities.

In the theory of planned behavior, several critical components are identified for social behaviors, namely, the attitude, the social norms, and the perceived behavior control. They affect the attentions or motivations which then in turn determine the actions. In this work, these factors have not been fully considered. The individuals are assumed implicitly to be in total control of switching among different distancing actions, which may not be true in general according to the studies on individual responses to public health measures [62, 63]. It would be interesting to take into account the influences on distancing decisions from the behavioral determinants suggested by the theory of planned behavior. For example, individuals with

different attitudes toward social distancing may be considered as different population groups of different risk perceptions; certain control factors such as the ability to participate in certain activities such as remote work may need to be considered before a distancing action can be carried out even if it is justified with the risk assessment.

Game theory has been applied to modeling health behaviors, most notably, by Bauch and Earn 2004 [116] for modeling the vaccination behaviors. By comparing the perceived risks from vaccination and infection, an individual decides with a certain probability to take the vaccine. If a sufficient portion of the population is already immune, either naturally or through vaccination, a self-interest individual would choose not vaccinated even if there is only a small risk associated with vaccination, a game played by the individuals against the population. Bauch and Earn 2004 show that a Nash equilibrium can be reached for the game with a vaccine uptake probability in between 0 and 1, which ultimately determines the vaccination level of the population. There is also a critical threshold for the vaccination level, beyond which vaccination can be avoided. The work in this paper is partially motivated by the success of the game model for vaccination by Bauch and Earn 2004.

Another interesting piece of game theoretic work on the health behaviors is done by Woike et al. 2022 [117]. An experimental game is carried out in [117] in two small but real populations, where the individuals are given choices to take low-risk, low-reward vs high-risk, high-reward protections from an assumed epidemic. It turns out that in several rounds of the game, a very high percentage of individuals tend to take the high-risk actions although highly likely to be infected and no rewards. The game is then conducted in several possible scenarios, where the players are provided with information on the development of the epidemic, the actions of other participants, healthcare promotions, outcomes of infections, etc. The results from the simulated game show how the individual actions are influenced by the provided information. While social distancing activities are complex and hard to survey, the simulated game in [117] can be quite helpful for not only providing insights into the individual behavioral responses to epidemics but also improving the existing epidemiological models including the model proposed in this work.

The games discussed in this work are assumed to have only a small number of very general social activities, i.e., 20 CASA activities. In practice, there can be many more activities. They can also be of more specific types. For example, there can be different workplaces, different restaurants, and different shopping centers, and going to each of these places may be considered as a different social activity. The activities may also have some connections, i.e., not necessarily be independent of each other. An individual who participates in one of the activities may have contacts with individuals in other activities when the activities are carried out in close proximity in time or space [90, 91].

The number of population groups in a heterogeneous population is not limited either, although only up to four population groups are considered in this work. The types of groups can also be combined. For example, groups can be formed according to the age as well as the vulnerability to the disease such as the vaccinated/unvaccinated children, recovered/unprotected adults, etc. In any case, the distancing activities in such populations can all be modeled as a multi-player game with each group corresponding to a single player and having its own risk assessment according to any conceivable social, economic, as well as health concerns.

In this work, the same logistic function $\sigma_i$ is used to define the risk function $p_i(y)$ with $\kappa_i = 10$ and $\theta_i = 0.5$ for all $i$. In practice, they may need to be specifically determined for specific activities, especially if the number and types of activities are to be expanded. The parameters $\delta_i$ do not need to be the same for all $i$ either. For example, in certain situations, close contacts in schools or workplaces are less of a concern, and $\delta_i$ for these activities may therefore be given a

smaller value than it is supposed to be. Further investigation into the use of these variations in practice can be pursued in future efforts.

When choosing among a group of activities, there must be personal constraints for individuals to attend some of them. For example, some people may have to go to hospital at least once a week; the essential workers or kindergarten teachers may have to go to work for a minimum amount of time. There must be certain time limits for some activities as well such as air traveling or even restaurant dinning. On the other hand, some individuals may never attend some of the activities. For example, children may never go to workplaces while seniors would hardly go to bars or schools. Such constraints have not been implemented in the current model in the paper. They should be included for the further expansion of the investigation in future.

The distancing game must have multiple equilibrium strategies in general, which have not been addressed in this work. This could happen when more realistic risk assessment functions are used, for example, when a different $\delta_i$ is set to a different activity or a different logistic function $\sigma_i$ is used for a different activity with different $\kappa_i$ and $\theta_i$ values. It could happen when constraints are introduced on the participating frequencies in some of the activities. It could also happen when some of the activities are dependent of each other as shown in previous studies [90, 91]. Social distancing is a complex social behavior and cannot be modeled successfully until all these factors are considered.

Finally, the social distancing behavior depends on the severity of the epidemic, while the latter is not constant and also depends on the former. For example, when the severity of infection is high, social distancing must be enhanced; when the severity of infection is low, social distancing may be relaxed. Such changes may happen back and forth in the entire epidemic time. They may in turn influence the dynamics of epidemics as well. Therefore, a complete model for social distancing must be coupled with an epidemiological model so that the changes of social distancing behaviors can be predicted successfully with changing epidemiological conditions and vice versa, as called for research on incorporating human behavior in epidemiological models in a recent US NSF program announcement [118].

## Methods

### Deriving contact and impact factors

The 20 CASA activities in Table 1 are grouped to 5 levels according to their possible contact rates: The lowest level—activities 1–4; the second level—activities 5–8; the third level—activities 9–12; the fourth level—activities 13–16; and the fifth level—activities 17–20. Consider a complete social distancing state when social distance is practiced by every individual to avoid close social contacts while maintaining only minimum social activities. An individual may stay home almost the whole day every day: he/she may spend 2 hours per day (or equivalently, 14 hours per week) for each of the first level activities; 1 hour per day (or equivalently, 7 hours per week) for each of the second; and so on and so forth, as listed in Table 2. Consider this strategy as an optimal strategy $x^*$ for every individual and hence $y^* = x^*$ for the population with $y_i^* =$

**Table 2. Assumed active times in 20 CASA activities for complete social distancing.**

| Activities: | 1 | 2 | 3 | 4 | 5 | 6 | 7 | 8 | 9 | 100 |
|---|---|---|---|---|---|---|---|---|---|---|
| Active time: | 14 | 14 | 14 | 14 | 7 | 7 | 7 | 7 | 4 | 4 |
| **Activities:** | **11** | **12** | **13** | **14** | **15** | **16** | **17** | **18** | **19** | **20** |
| Active time: | 4 | 4 | 2 | 2 | 2 | 2 | 1 | 1 | 1 | 1 |

Activities—the same as in Table 1; Active time—active time in hours per week (112 active hours)

**Table 3. Assumed active times in 20 CASA activities when free of social distancing.**

| Activities: | 1 | 2 | 3 | 4 | 5 | 6 | 7 | 8 | 9 | 10 |
|---|---|---|---|---|---|---|---|---|---|---|
| Active time: | 5 | 5 | 3 | 3 | 5 | 3 | 3 | 5 | 5 | 3 |
| Activities: | 11 | 12 | 13 | 14 | 15 | 16 | 17 | 18 | 19 | 20 |
| Active time: | 3 | 5 | 4 | 4 | 16 | 20 | 6 | 6 | 4 | 4 |

Activities—the same as in Table 1; Active time—active time in hours per week (112 active hours)

$x_i^* =$ (active time in $i$)/112 as the participating frequency in activity $i$. Then, based on general theory for population games [58, 59], there must be a constant $\lambda$ such that $p_i(y^*) = \lambda$ for all $i$ such that $y_i^* > 0$. It follows that $w_i \sigma_i(y_i^*) = \lambda$ for all $i$. Set $\lambda = \sigma_1(y_1^*)$. Then, $w_i = \lambda / \sigma_i(y_i^*)$. Let $\alpha_i = w_i$ for all $i$. The contact factors $\alpha_i$ are obtained as listed in Table 1.

Similarly, consider a population state free of social distancing, i.e., the individuals are free to choose their activities as if there is no epidemics and no social distancing: An individual may spend only 1 hours per day (or total 5 hours per week) for reading or watching TV; go to movie theaters 4 times (total 4 hours per week); work at home only 1 hour per day (or total 5 hours per week); and go to workplaces for 5 hours a day (or total 20 hours per week), etc. as listed in Table 3. Consider this strategy as an optimal strategy $x^*$ for every individual and hence $y^* = x^*$ for the population with $y_i^* = x_i^* =$ (active time in $i$)/112 as the participating frequency in activity $i$. Then, based on general theory for population games [58, 59], there must be a constant $\lambda$ such that $p_i(y^*) = \lambda$ for all $i$ such that $y_i^* > 0$. It follows that $w_i \sigma_i(y_i^*) = \lambda$ for all $i$. Keep $\lambda$ to be at the same level as in the complete social distancing state. Then, $w_i = \lambda / \sigma_i(y_i^*)$. Let $\beta_i = w_i$ for all $i$. The impact factors $\beta_i$ are obtained as listed in Table 1.

## Equilibrium strategies and stabilities

For each activity $i$, $p_i(y) = w_i \sigma_i(y_i)$ with $w_i = \delta_i \alpha_i + (1 - \delta_i) \beta_i$ and $0 \leq \delta_i \leq 1$. If $w_i > 0$ for all $i$, the game reaches equilibrium when the potential distancing risks at all the activities are the same, i.e., an optimal strategy $x^*$ and hence $y^*$ is found such that $p_i(y^*) = w_i \sigma_i(y_i^*) = \lambda$ for all $i$ for some constant $\lambda$, assuming $x_i^* > 0$ for all $i$. It follows that $y_i^* = \sigma_i^{-1}(\lambda / w_i)$ with $\Sigma_i \sigma_i^{-1}(\lambda / w_i) = 1$ (proofs in **Appendix—General distancing games**).

Note also that an equilibrium strategy $x^*$, $x_i^* > 0$ for all $i$ and hence $y^*$, $y_i^* > 0$ for all $i$ is evolutionarily stable—a term used in evolutionary game theory [58, 59]. It means that if there is a small change in the strategy, the equilibrium strategy still prevails. In other words, if the population strategy $y^*$ is perturbed (or invaded) slightly by a new strategy $y$, $y^*$ will remain to be a better choice than $y$, and not be taken over by $y$ (proofs in **Appendix—Evolutionary stability**).

## Generating small-world social networks

The social network is generated with the well-known Watts-Strogatz algorithm [104]. The algorithm has three parameters to determine a social network, $m$ the number of the nodes, $K$ the average degree of the nodes, and $b$ the randomness of the connections. The algorithm generates a small-world social network of $m$ nodes for a population of $m$ individuals. In a cyclic order, the algorithm first connects each node with $K/2$ nodes next to the node on the right and then on the left. Then, for each node $i$ and node $j$ of $K/2$ nodes connected to node $i$ on the right, the algorithm selects a node $k$ not connected to $i$, and with a probability $b$, removes the link between $i$ and $j$ and connects $i$ and $k$. In this way, the average degree of the nodes in the network would be around $K$, and the randomness of the connection between the connected

nodes can be specified by *b*. A detailed algorithmic description for generating a small-world social network is given in Algorithm 1. A Matlab code can be found in the provided S1–S3 Files.

**Algorithm 1** Generate a small-world network: (*m*, *K*, *b*)

```
Require: m integer ∨ K even ∨ 0 ≤ b ≤ 1
Ensure: Set m nodes in a cycling order
1: For i = 1: m
2:   Connect i with K/2 nodes on its left
3:   Connect i with K/2 nodes on its right
4: End
5: For i = 1: m
6:   For each j of K/2 nodes next to i on the right
7:     If (i, j) connected
8:       Find a node k not connected with i
9:       Disconnect (i, j) and connect (i, k) with a probability b
10:    End
11:  End
12: End
```

## Simulation of distancing games in social networks

The simulation of distancing games is based on the general principle of replicator dynamics for population games [58, 59]. If $y_i$ is the participating frequency of the population in activity *i* at a certain time *t*, the replicator dynamics states that the changing rate of $y_i$ is proportional to the difference between the potential distancing risk $p_i(y)$ at activity *i* and the population average $\Sigma_i y_i p_i(y)$. If the potential distancing risk is higher than the average, activity *i* is considered to be riskier, and $y_i$ (and hence $x_i$) should be decreased; otherwise, $y_i$ (and hence $x_i$) should be increased.

For the game on a social network, the simulation can be done for every individual against the population in his/her neighborhood. The population strategy *y* then becomes the neighborhood strategy, which is different for a different individual in general. The neighborhood of size *k* of an individual includes all the neighbors connected to the individual with up to *k* consecutive links. Depending on the neighborhood size, the game becomes against a fraction of population ranging from the immediate neighbors of the individual to the whole population. Algorithm 2 gives more algorithmic description on how an individual updates his/her strategy in every round of the game. A Matlab code for the whole simulation is provided in S1 File.

The simulation starts with a strategy for every individual randomly generated around its equilibrium one. More specifically, if $x_i^*$ is the equilibrium value of the participating frequency for activity *i*, then first perturb $x_i^*$ randomly by a certain percentage $\rho$, say $\rho = 20\%$, and then generate $x_i$ randomly within 100% of deviation from the perturbed value of $x_i^*$. The simulation proceeds in multiple generations. At each generation, every individual plays the game once, i.e., has a chance to update his/her strategy. The simulation ends when either every individual strategy converges to the equilibrium strategy in average or it stops making any further progress. Every simulation is repeated for 5 times and an average output is recorded and reported.

**Algorithm 2** Updating individual distancing strategies

```
Require: Individual strategy x, neighborhood strategy y
1: p̄_y = Σ_i y_i p_i(y)
2: For i = 1: n
3:   If p_i(y) < p̄_y
4:     If x_i < y_i
5:       x_i = x_i + 0.9 × (y_i − x_i)
6:     Else
```

```
 7:        x_i = x_i + 0.1 × min{1 - x_i, x_i - y_i}
 8:      End
 9:    End
10:    If p_i(y) > p̄_y
11:      If x_i > y_i
12:        x_i = x_i - 0.9 × (x_i - y_i)
13:      Else
14:        x_i = x_i - 0.1 × min{x_i, y_i - x_i}
15:      End
16:    End
17:    If |p_i(y) - p̄_y| < 0.01
18:      If x_i > y_i
19:        x_i = x_i - 0.5 × (x_i - y_i)
20:      End
21:      If x_i < y_i
22:        x_i = x_i + 0.5 × (x_i - y_i)
23:      End
24:    End
25: End
26: x = x/Σ_i x_i
```

## Distancing in heterogeneous populations

Consider a simple case where the population is divided into two groups, groups $a$ and $b$. Let $x^a$ and $x^b$ be the distancing strategies for individuals in groups $a$ and $b$, respectively, with $x_i^a$ and $x_i^b$ being the participating frequencies of the individuals in activity $i$. Let $y^a$ and $y^b$ be the average strategies of the individuals in groups $a$ and $b$ in the population, with $y_i^a$ and $y_i^b$ being the corresponding average participating frequencies of these individuals in activity $i$. Given strategies $y^a$ and $y^b$ in the population, the potential distancing risk at activity $i$ can be estimated by a function $p_i^a(y^a, y^b) = (1 + s)w_i^a\sigma_i(y_i^a) + w_i^b\sigma_i(y_i^b)$ for a group $a$ individual or $p_i^b(y^a, y^b) = w_i^a\sigma_i(y_i^a) + (1 + s)w_i^b\sigma_i(y_i^b)$ for a group $b$ individual, where $w_i^a$ and $w_i^b$ are risk factors for groups $a$ and $b$, respectively, and the contribution to the distancing risk from the individual's own group is given more weight $(1 + s)$ for some $s > 0$, as the individual is supposed to interact more with the individuals in his/her own group than those in the other group.

Then, for an individual of strategy $x^a$ in group $a$, the distancing risk to participate in given $n$ activities can be evaluated by a function $\pi_a(x^a, y^a, y^b) = \Sigma_i x_i^a p_i^a(y^a, y^b)$. Similarly, for an individual of strategy $x^b$ in group $b$, the distancing risk to participate in given $n$ activities can be evaluated by a function $\pi_b(x^b, y^a, y^b) = \Sigma_i x_i^b p_i^b(y^a, y^b)$. Together, with these functions, a multi-player distancing game can be defined for the whole population with each population group corresponding to a single player; and a pair of strategies $x^{a*}$ and $x^{b*}$ form an equilibrium pair of strategies for the game if and only if $y^{a*} = x^{a*}$ and $y^{b*} = x^{b*}$, and for every individual in group $a$, the distancing risk $\pi_a(x^{a*}, y^{a*}, y^{b*})$ using strategy $x^{a*}$ is no greater than the distancing risk $\pi_a(x^a, y^{a*}, y^{b*})$ using any other strategy $x^a$, and for every individual in group $b$, the distancing risk $\pi_b(x^{b*}, y^{a*}, y^{b*})$ using strategy $x^{b*}$ is no greater than the distancing risk $\pi_b(x^b, y^{a*}, y^{b*})$ using any other strategy $x^b$.

It follows from a little bit analysis that the equilibrium strategy for each population group can be obtained with $x_i^{a*} = \sigma_i^{-1}(\lambda_a/w_i^a)$ for all $i$ for some constant $\lambda_a$ such that $\Sigma_i \sigma_i^{-1}(\lambda_a/w_i^a) = 1$, and $x_i^{b*} = \sigma_i^{-1}(\lambda_b/w_i^b)$ for all $i$ for some constant $\lambda_b$ such that $\Sigma_i \sigma_i^{-1}(\lambda_b/w_i^b) = 1$, assuming $x_i^{a*} > 0$ and $x_i^{b*} > 0$ for all $i$. At equilibrium, $y^{a*} = x^{a*}$ and $y^{b*} = x^{b*}$, and the average population strategy should be $y^* = \rho_a y^{a*} + \rho_b y^{b*}$, where $\rho_a$ and $\rho_b$ are the percentages of group $a$ and $b$ individuals in the population, respectively. These results can be

extended straightforwardly to populations with more than two population groups (general descriptions and proofs in **Appendix—Games with multiple population groups**).

The simulation of the distancing game with multiple population groups is done with the population also distributed over a small-world social network, where the game is played by every individual against his/her population group in his/her neighborhood. The key difference of this simulation from the one described in Algorithm 2 is that the potential distancing risk at each activity is estimated using a formula as described above, and the average potential distancing risk over all activities is evaluated for each individual using his/her group strategy in his/her neighborhood. A Matlab code for simulating the distancing games with up to four population groups is provided in S2 File.

## Following the leaders vs. following the crowd

A certain percentage of individuals are randomly selected as leaders. A leader makes distancing decisions as a regular player for the distancing game, whether the game is played in a small-world social network or with multiple population groups. A follower tries to find the leaders in his/her population group in his/her neighborhood, and copies the average strategy of the leaders. If he/she fails to find a leader among his/her closest neighbors, he/she either makes her own decision as a regular player or follows the crowd by copying the average strategy of his/her group members in his/her neighborhood. The simulation is done with the percentage of leaders in the population varying from 10% to 20%, 30%, 40%, and 50%, and the neighborhood size changing from 1 to 2, 3, 4, 5, and 6. The parameters for the network are fixed to $m = 2000$, $K = 6$, and $b = 0.3$. A Matlab code for the simulation is provided in S3 File.

## Research ethics statement

This work does not involve any human subjects. There is no primary and secondary data on human subjects collected. The work is conducted to develop and contribute to generalizable scientific knowledge. The work ethics oversight is waived by the Iowa State University IRB Office, with the following written statement:

From: IRBManager on behalf of ISU IRB Administrator

To: Zhijun Wu

Subject: Beyond six feet: The collective behavior of social distancing

Your responses on the Human Subjects Research Assessment form (Does My Study Require IRB Oversight) indicate that your project does not involve research per the federal regulations (45CFR46.102 and 21CFR56). Accordingly, IRB oversight is not necessary.

https://compliance.iastate.edu/research-ethics-compliance/irb/

## Notes on simulation results

All results from computer simulation conducted in this work are documented in Supplementary Information. The Matlab codes producing the results are all provided, including a set of codes for producing the plots in the paper. To run the codes, the pdf files need to be converted into text files with.m extensions. The code descriptions can be found at the beginning of the files.

## Appendix

### General distancing games

Assume that the population has $n$ activities. Let $x = \{x_i: i = 1, \ldots, n\}$ be a set of frequencies representing the distancing strategy of any individual, with $x_i$ being the frequency of the

individual to participate in activity $i$, and $\Sigma_i \, x_i = 1$. Let $y = \{y_i : i = 1, \ldots, n\}$ be a set of frequencies representing the strategy of the population, with $y_i$ being the average frequency of all the individuals in the population to participate in activity $i$, and $\Sigma_i \, y_i = 1$.

Given a distancing strategy $y$ from the population, assume that each individual can estimate the potential distancing risk at each activity $i$ using a function $p_i(y)$. Then, the distancing risk of the individual of strategy $x$ at activity $i$ must be $x_i p_i(y)$, and at all the activities together be $\Sigma_i \, x_i p_i(y) = \pi(x, y)$.

**Definition 1** (Distancing game). *A distancing game is a population game where every individual chooses a strategy $x$ against a strategy $y$ of the population so that his/her distancing risk $\pi(x, y)$ can be minimized.*

**Definition 2** (Equilibrium strategy). *A strategy $x^*$ is an equilibrium strategy of the distancing game if and only if every individual in the population takes this strategy $x^*$ (and hence $y^* = x^*$) such that his/her distancing risk $\pi(x^*, y^*) \leq \pi(x, y^*)$ for any strategy $x$.*

**Theorem 1**. *A strategy $x^*$ is an equilibrium strategy for the distancing game if and only if there is a constant $\lambda$ such that*

$$x_i^*(p_i(y^*) - \lambda) = 0, \quad x_i^* \geq 0, \quad p_i(y^*) - \lambda \geq 0, \quad i = 1, \ldots, n. \tag{1}$$

*Proof.* ($=>$) Suppose that $x^*$ is an equilibrium strategy. Then, $\pi(x^*, y^*) \leq \pi(x, y^*)$ for any strategy $x$, and therefore, $\pi(x^*, y^*) \leq \pi(e_i, y^*) = p_i(y^*)$, $i = 1, \ldots, n$, where $e_i$ is the $i$th unit vector. Let $\pi(x^*, y^*) = \lambda$. Then $p_i(y^*) - \lambda \geq 0$ for all $i = 1, \ldots, n$. For any $i$, if $x_i^* = 0$, $x_i^*(p_i(y^*) - \lambda) = 0$; if $x_i^* > 0$, $(p_i(y^*) - \lambda)$ must be zero, for otherwise, $x_i^*(p_i(y^*) - \lambda) > 0$. Collect the latter inequality for all $i$ to obtain $\Sigma_i \, x_i^*(p_i(y^*) - \lambda) > 0$. Then $\pi(x^*, y^*) - \lambda > 0$, which is contradictory to the fact that $\pi(x^*, y^*) = \lambda$. Thus the conditions in (1) are all satisfied.

($<=$) Suppose there is a parameter $\lambda$ such that $x^*$ satisfies all the conditions in (1). Collect the first equation in (1) for all $i$ to obtain $\Sigma_i \, x_i^*(p_i(y^*) - \lambda) = 0$, which is equivalent to $\pi(x^*, y^*) - \lambda = 0$, and therefore, $\lambda = \pi(x^*, y^*)$. Let $x$ be an arbitrary strategy. Multiply the last equation in (1) by $x_i$ to obtain $x_i(p_i(y^*) - \lambda) \geq 0$. Collect the latter inequality to obtain $\Sigma_i \, x_i(p_i(y^*) - \lambda) \geq 0$, which is equivalent to $\pi(x, y^*) - \lambda \geq 0$. Then, $\pi(x^*, y^*) \leq \pi(x, y^*)$ for any strategy $x$, and $x^*$ is an equilibrium strategy.

**Theorem 2**. *Assume that the activities are independent and function $p_i(y) = w_i \sigma_i(y_i)$ with $w_i > 0$ for all $i$. Then, $x_i^* = \sigma_i^{-1}(\lambda/w_i)$, $i = 1, \ldots, n$, form an equilibrium strategy for the distancing game, assuming $x_i^* > 0$ for all $i$, where $\lambda$ is a constant such that $\Sigma_i \, \sigma_i^{-1}(\lambda/w_i) = 1$.*

*Proof.* By Theorem 1, since $x_i^* > 0$ for all $i$, there is a constant $\lambda$ such that $p_i(y^*) - \lambda = 0$ for all $i$. It follows that $w_i \sigma_i(y_i^*) = w_i \sigma_i(x_i^*) = \lambda$, and $x_i^* = \sigma_i^{-1}(\lambda/w_i)$. Since the sum of all $x_i^*$ equals 1, the sum of the latter equations gives $\Sigma_i \, \sigma_i^{-1}(\lambda/w_i) = 1$.

## Evolutionary stability

**Definition 3** (Evolutionary stability). *An equilibrium strategy $x^*$ for the distancing game is evolutionarily stable if for any strategy $x \neq x^*$, there is $\bar{\epsilon} \in (0, 1)$ such that $\pi(x^*, \epsilon x + (1 - \epsilon)x^*) < \pi(x, \epsilon x + (1 - \epsilon)x^*)$ for all $\epsilon \in (0, \bar{\epsilon})$ [58, 59].*

**Definition 4** (Potential minimization). *Let $f(y)$ be a function such that $f_{y_i}(y) = p_i(y)$, $i = 1, \ldots, n$. Then, the problem $\min\{f(y): \Sigma_i \, y_i = 1, y_i \geq 0, i = 1, \ldots, n\}$ is called a potential minimization problem for the distancing game defined by $p_i(y)$, $i = 1, \ldots, n$.*

**Theorem 3**. *A strategy $x^*$ is an equilibrium strategy for the distancing game if and only if $x^*$ is a KKT point of the corresponding potential minimization problem [119, 120].*

**Theorem 4**. *An equilibrium strategy $x^*$ for the distancing game is evolutionarily stable if and only if $x^*$ is a strict local minimizer of the corresponding potential minimization problem [119, 120].*

**Theorem 5**. *Assume that the activities are independent and function $p_i(y) = w_i\sigma_i(y_i)$ with $w_i > 0$ for all i. Then, the equilibrium strategy $x_i^* = \sigma_i^{-1}(\lambda/w_i)$ with $\Sigma_i\, \sigma_i^{-1}(\lambda/w_i) = 1$ for the distancing game, assuming $x_i^* > 0$ for all i, is evolutionarily stable.*

*Proof.* The Hessian of the objective function *f(x)* of the potential minimization problem corresponding to the distancing game is a diagonal matrix with $w_i\sigma_i'(y_i^*)$, $i = 1, \ldots, n$ as the diagonal elements. Since $w_i\sigma_i'(y_i^*) > 0$ for all *i*, the Hessian is positive definite, which guarantees the solution to the potential minimization problem *x** to be a strict local minimizer. It follows that *x** must be evolutionarily stable by Theorem 4.

## Games with multiple population groups

Assume that the population is divided into *M* groups. Let $x^{(j)}$ be the strategy of an individual in group *j*, and $y^{(j)}$ the average strategy of all group *j* individuals in the population. Let $w_i^{(j)}$ be the risk factor for activity *i* for the individuals in group *j*. Then, the potential distancing risk for a group *j* individual at activity *i* can be defined as

$$p_i^{(j)}(y^{(j)}, y^{(-j)}) = \Sigma_k\, s_{jk}\, w_i^{(k)}\sigma_i(y_i^{(k)}) \tag{2}$$

where $y^{(-j)}$ represents all group strategies $y^{(1)}, \ldots, y^{(M)}$ excluding $y^{(j)}$, $\Sigma_k$ means the sum over all $k = 1, \ldots, M$, and $s_{jk}$ is a scaling factor, $s_{jk} = (1 + s)$ for some $s > 0$ if $k = j$ and $s_{jk} = 1$ if $k \neq j$, thus the contribution of group *j* to the distancing risk is given more weight as an individual in group *j* is supposed to interact more with the individuals in his/her own group than those in other groups.

**Definition 5** (Distancing game with multiple population groups). *Assume that the population is divided into M groups. Let $\pi_j(x^{(j)}, y^{(j)}, y^{(-j)}) = \Sigma_i\, x_i^{(j)} p_i^{(j)}(y^{(j)}, y^{(-j)})$ be the distancing risk of the individual of strategy $x^{(j)}$ in group j, $j = 1, \ldots, M$. Then together with all these functions, a multi-player distancing game can be formed with each population group corresponding to a single player; and a set of strategies $x^{(j)*}$, $j = 1, \ldots, M$, is an equilibrium set of strategies for the game if and only if for all $j = 1, \ldots, M$, $y^{(j)*} = x^{(j)*}$, and $\pi_j(x^{(j)*}, y^{(j)*}, y^{(-j)*}) \leq \pi_j(x^{(j)}, y^{(j)*}, y^{(-j)*})$ for any strategy $x^{(j)}$.*

**Theorem 6**. *Assume that the population is divided into M groups and the activities are independent. Assume that the function for an individual in group j to evaluate the potential distancing risk at activity i is given by (2) with $w_i^{(j)} > 0$ for all i and j. Then, there is a unique set of equilibrium strategies $x^{(j)*}$, $j = 1, \ldots, M$, for the multi-player distancing game of the population, with $x_i^{(j)*} = \sigma_i^{-1}(\lambda^{(j)}/w_i^{(j)})$ for all i, assuming $x_i^{(j)*} > 0$ for all i and j, where $\lambda^{(j)}$ is a constant such that $\Sigma_i\, \sigma_i^{-1}(\lambda^{(j)}/w_i^{(j)}) = 1$.*

*Proof.* Let $x^{(j)*}$ be the strategy of an individual in group *j* at equilibrium and $y^{(j)*}$ the average strategy of group *j* individuals in the population. Then, it is necessary and sufficient that for each group *j*, $p_i^{(j)}(y^{(j)*}, y^{(-j)*}) = t_j$ for all *i* for some constant $t_j$, i.e.,

$$s_{j1}w_i^{(1)}\sigma_i(y_i^{(1)*}) + \cdots + s_{jj}w_i^{(j)}\sigma_i(y_i^{(j)*}) + \cdots + s_{jM}w_i^{(M)}\sigma_i(y_i^{(M)*}) = t_j$$
$$i = 1, \ldots, n; \quad j = 1, \ldots, M. \tag{3}$$

The above equations can be written in a more compact form as:

$$Sz_i = t, \quad i = 1, \ldots, n \tag{4}$$

where $S = (s_{jk})$ is an $M \times M$ matrix, $s_{jk} = (1 + s)$ for some $s > 0$ if $j = k$ and $s_{jk} = 1$ if $j \neq k$, $z_i$ and *t* are *M*-vectors, $z_i^T = (w_i^{(1)}\sigma_i(y_i^{(1)*}), \ldots, w_i^{(M)}\sigma_i(y_i^{(M)*}))$, and $t^T = (t_1, \ldots, t_M)$. It is not difficult to

verify that $S$ is nonsingular. Therefore, $z_i = S^{-1}t$. Let $(S^{-1}t)_j = \lambda^{(j)}$, $j = 1, \ldots, M$. Then, $w_i^{(j)}\sigma_i(y_i^{(j)*}) = \lambda^{(j)}$ for all $i$. It follows that $y_i^{(j)*} = \sigma_i^{-1}(\lambda^{(j)}/w_i^{(j)})$ with $1 = \Sigma_i\,\sigma_i^{-1}(\lambda^{(j)}/w_i^{(j)})$.

## Supporting information

**S1 Text.**
(PDF)

**S2 Text.**
(PDF)

**S3 Text.**
(PDF)

**S1 File.**
(PDF)

**S2 File.**
(PDF)

**S3 File.**
(PDF)

**S1 Fig.**
(PDF)

## Acknowledgments

The author would also like to thank the anonymous reviewers for carefully reading the manuscript and providing valuable comments and suggestions for revising and improving the paper.

## Author Contributions

**Conceptualization:** Zhijun Wu.

**Data curation:** Zhijun Wu.

**Formal analysis:** Zhijun Wu.

**Funding acquisition:** Zhijun Wu.

**Investigation:** Zhijun Wu.

**Methodology:** Zhijun Wu.

**Project administration:** Zhijun Wu.

**Resources:** Zhijun Wu.

**Software:** Zhijun Wu.

**Supervision:** Zhijun Wu.

**Validation:** Zhijun Wu.

**Visualization:** Zhijun Wu.

**Writing – original draft:** Zhijun Wu.

**Writing – review & editing:** Zhijun Wu.

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
