## [Decision Letter · Decision Letter 0]

22 Feb 2024

PONE-D-23-33255Beyond six feet:  The collective behavior of social distancingPLOS ONE

Dear Dr. Wu,

Thank you for submitting your manuscript to PLOS ONE. After careful consideration, we feel that it has merit but does not fully meet PLOS ONE’s publication criteria as it currently stands. Therefore, we invite you to submit a revised version of the manuscript that addresses the points raised during the review process.

The first reviewer is overall positive about your manuscript, but raises various points that need to be addressed. In particular, concerns about theoretical foundations and justifications for certain assumptions need to be addressed. This aligns with comments by the second reviewer, who raises serious concerns about the assumptions underpinning your game theoretical model. In particular, assumptions about the potential distancing risk p_i(y) need to be addressed. The reviewer makes a strong case for rejecting the manuscript based on these comments, but I want to provide you the opportunity to respond and revise you modelling assumptions in response. If you decide to resubmit a revised version, please ensure you have addressed these concerns and have revised your assumptions appropriately.

We look forward to receiving your revised manuscript.

Kind regards,

Vincent Antonio Traag, Ph.D.

Academic Editor

PLOS ONE

Journal Requirements:

"Work supported partially by the Simons Foundation Mathematics and Physical Sciences Collaboration Grants for Mathematicians (Award Number: 586065)."

5. We are unable to open your Supporting Information file SI-simulation-code-1.zip, SI-simulation-code-2.zip and SI-simulation-code-3.zip. Please kindly revise as necessary and re-upload.

Reviewers' comments:

Reviewer's Responses to Questions

**Comments to the Author**

1. Is the manuscript technically sound, and do the data support the conclusions?

Reviewer #1: Yes

Reviewer #2: Partly

2. Has the statistical analysis been performed appropriately and rigorously? 

Reviewer #1: Yes

Reviewer #2: Yes

3. Have the authors made all data underlying the findings in their manuscript fully available?

Reviewer #1: Yes

Reviewer #2: Yes

4. Is the manuscript presented in an intelligible fashion and written in standard English?

Reviewer #1: Yes

Reviewer #2: Yes

5. Review Comments to the Author

Reviewer #1: The manuscript “Beyond six feet: The collective behavior of social distancing” presents a game-theoretic model for analyzing and simulating distancing strategies during infectious disease outbreaks. The model introduced is sound; the research has been conducted comprehensively, thoroughly, and systematically; the results are supported by detailed mathematical proofs and derivations as well as detailed descriptions of the simulations; and the manuscript is well-structured and well-written. The findings give interesting and valuable insights into the modeling of distancing strategies in game-theoretic scenarios of disease spread. The approach of starting with a highly abstract, simplified version of the model and then gradually introducing more complexity and details stands out to me, shows the rigor of the author, and enables me to understand the increasingly complex dynamics.

However, there are some important aspects that, I think, need to be addressed, before the manuscript can be published. My main concerns are:

1: In the current version of the manuscript, the model lacks a theoretical foundation in the behavioral sciences. However, without proper grounding in behavioral theory, this research remains a modeling exercise.

Starting points for health behavioral theories that consider risk perception could be the “Theory of Planned Behavior” (Godin & Kok, 1996) or the Health Belief Model (Green & Murphy, 2014). Empirical support could be found in reviews by Bish & Michie (2010) and Leppin & Aro (2009). Another starting point, to highlight the importance of considering risk perception in infectious disease modeling, could be reference 45 in the presented manuscript.

2: In addition to the previous comment, the author claims that “Work on modeling social distancing [… had] little specifics on how the distancing activities are carried out and how certain distancing patterns or levels are achieved.” I would agree that this is an emerging field. However, there is recent literature that can be referenced here. For example, Nunner et al. (2022) showed in a model study with risk perception that clusters of individuals that reduce social contacts homogeneously can mitigate disease spread. The model is based on a game-theoretic model of network formation by Jackson (2008) with comparable considerations (as described in lines 42-49). Another example would be Woike et al. (2022), who explore the effectiveness of behavioral interventions in controlling the spread of infectious diseases through simulations and experiments. A comparison of the results with the findings in these studies would provide an interesting extension of the discussion.

3. The discussion is mostly a (well-written) summary of the paper. However, it lacks a more explicit discussion of the results in the context of the literature and the practical implications of the research for epidemic intervention and public health policy. Some previous comments might be useful for a more refined discussion.

4. Presentation of the results regarding tables and figures should be improved.

(a) I find Table 1 difficult to read. A vertical layout with activities written out would improve accessibility.

(b) Table 2 shows the same information as Figure 1, while Figure 1 is more accessible. Furthermore, Figure 1 should not use lines because they suggest continuous data on the x-axis. A plot in the style of Figure 3 would be more fitting.

(c) Tables 3 and 4 might be difficult to access for a broader interdisciplinary audience, such as the readership of PLOS ONE. Providing some more intuition about the Euclidian norm would help.

Suggestion: A plot that shows the combination of results for all scenarios, in the style of Figure 3, would enable a better and more intuitive comparison of the results across the different scenarios and relax the issues above.

5. Why was k = 3 chosen for the networks in the heterogeneous and leader-follower scenarios? Earlier (line 207), a setting of k = 6 was supported by literature. Although the Milgram experiment has received much criticism over the years, it provides reasoning. Context-related research, such as the POLYMOD study by Mossong et al. (2008) or similar work on social encounter networks in Great Britain by Danon (2013), is more suitable to defend degree settings. The former would also provide more solid empirical footing for the decision to distinguish contact numbers by age.

6. The “following the leaders” parts (Results and Methods) require refinement and further elaboration:

(a) "A small number of leaders" is stated multiple times (e.g., line 90, line 289). However, this number can actually go up to 60% (line 561).

(b) How were the leaders selected?

(c) How far can followers be from leaders, while still considering them to be leaders? (line 555-557)

(d) What are the “multiple population groups” (line 555), how are they defined, and how large are they?

(e) How is it decided whether an individual makes his/her own decision as a regular player or follows the crowd (lines 558-559)?

Other concerns are:

1: References 27-51 (lines 7-10) are collectively used within a single sentence to cover a wide range of topics related to social distancing, including the impact on mental health, economic considerations, and strategies for optimizing social activities. This does not pay sufficient tribute to the literature. A more refined discussion on the different aspects would provide a clearer and more focused analysis of the various aspects of social distancing, and thus provide a better justification of the research presented.

2: Line 21-22: “The decision of each individual must depend on or be influenced by the actions of all other individuals in the population. For example, if everybody decides to stay home, an individual may choose to watch a movie although the risk of having close contacts at a movie theater is high”

I don’t quite understand, why the decision “must” depend on or be influenced by the actions of “all other individuals”. Is this only in the context of the model? A decision to watch a movie at home alone can be made just because I feel like doing it, and is thus independent of whether “all the people I know” do the same or something else. Furthermore, I don’t understand the example. If everybody stays home and one person goes to the theater, the risk of getting infected is low.

3. I understand that for modeling purposes, things have to be simplified. However, as justification for the leader-follower scenario, I consider the statement, “Indeed, in practice, it is most likely that only a small number of individuals such as public health experts or community leaders make decisions or recommendations while all others just follow.” (lines 89-91) a bold claim. I believe that there are more factors at play and much more heterogeneity among followers of individuals.

4. The same applies to “It shows how the distancing game is played out in possible realistic as well as idealistic scenarios, […]” (lines 93-94). Claiming that a simulated scenario is “realistic” is a little too far. Simulations always use abstraction.

5. To me, the term “negative social/economic impacts” (e.g., line 123) lacks intuition and seems strange to be combined. How do these combined negative impacts come about? Something can have a positive social but negative economic impact (e.g., inviting a group of friends to an expensive restaurant).

6. It would be interesting to learn whether the times required for convergence (lines 210-214) differ depending on k. Furthermore, I would encourage the author to have a more detailed discussion of these results (i.e., what are the implications of this finding).

7. The statements made in lines 219-221 and especially in lines 223-225 require references for backup.

8. How were the 20 CASA activities selected? The reference to the CDC did not prove useful when I tried to reconstruct a rationale for the explicitly selected activities. Furthermore, the description of contact factors determined lacks details on methodology and data sources ("according to common practices and CDC recommendations", line 423).

9. Reference 52 calls for “spatial distancing and social closeness: not for social distancing!”. Among many epidemiologists, “physical distancing” is now considered a more fitting term. I agree with the concerns raised regarding the term “social distancing” and would therefore encourage the author to reconsider the choice of terminology.

Reviewer #2: Dear Author,

I am extremely sympathetic to your efforts in devising adequate policy recommendations that can help deal with future epidemics. Adjusting policies can help deal with these epidemics in a better way that what was done for the COVID-19 crisis. This can be of much value to society as a whole. However any mistake can be extremely costly and potentially fatal to individuals. So any recommendations on this topic should be strongly backed up by serious and sound analysis. While I do not doubt the seriousness of your efforts, I am not convinced by the analysis for several reasons. I believe that the simplifying assumptions (which are not discussed) available in the supplementary material, "Methods" section, are too restrictive and eliminate a number of potentially quite relevant effects.

I understand that the simulations you offer could be redone with a more precise computation of the multipliers (alpha and beta) per activity, as you mention. Providing a preliminary simulation is valuable, provided that it is made clear that these results are illustratory in nature and should not be the basis of recommendations. You do mention this before presenting the table with estimates. So this is not where the issue lays, but in the underlying model of the population game.

The core issues are with the specification of the population game and the fact that infection risk and economic benefits get added up. The remainder of the analysis, using networks and population subgroups is quite interesting and should have been a real "plus". But all the results obtained are simulated on the basis of an equilibrium which is highly unrealistic and unsatisfactory to describe an epidemic.

(The result of evolutionary stability applies only for infinitesimal perturbations to a player's strategy in the population game, and certainly not to changes in strategies in the face of shocks such as a major epidemic.)

1. The simplifying assumptions in the model (cf. supplementary material) turn every distancing effect into a linear one. So the author adds linear effects, which is both quite unrealistic and mathematically problematic in the context of the study. It is highly unlikely that efforts and consequences of these efforts are linear. Moreover the linearity of all these effects lead to very specific equilibria, which are presumably quite different from the ones one would have if at least one effect was non linear. In more detail:

A major element of the computation, function p_i(y) is simply said to be (line 109, p 3) the "potential distancing risk of having close social contacts and negative social/economic impacts in activity i, when fully participated". This compounds social and economic impacts, that can be very different and potentially require different policy interventions. At the very least, this function should be discussed and explained. The function is assumed, for each activity, to be composed of two parts, a contact rate and an economic part. *But* both are assumed to be proportional only to the participating frequency y_i of the population in activity i. So everything is linear / additive. While the interpretation of the multiplier for contact rates and for social/economic aspects are different, the total effect is a sum linear in participating frequency. pi(y) = delta_i alpha_i y_i +(1 – delta_i) beta_i_iyi = w_i y_i

This is highly unsatisfactory as it eliminates numerous potential effects of non linearity: It may very well be the case that in some activities, a minimal threshold of participation is necessary and / or sufficient to maintain a large part of the economic gains, or a large part of the social gains. For instance, social interactions that must be frequent enough to reach a level of friendship or a personal wellbeing. Economic activity participation is non continuous due to labor market constraints on work duration and also the need to be visible to clients and employers. This would lead to completely different strategies, where individuals could chose to be at exactly the threshold for a number of activities and at some level close to 0 for others. This appears to be a much more realistic description of the way people behaved during the COVID-19 crisis. This is important to understand as it means that more efforts should be put in providing adequate protection from the epidemics in some activities that both have threshold values to obtain benefits and have high benefits to individuals. For instance invest in distancing widgets or remote technologies to provide these benefits for these activities, and not invest in others where such discontinuous effects do not exist.

2. An obvious additional non-linearity is the one that arises from budget / subsistence constraints : Many people need to work in order to meet their subsistence requirements. The strength of this constraint and the number of people strongly affected by it depend crucially on regional characteristics pre-epidemic but also on policy responses to the epidemic.

3. More generally in many situations gains are concave (diminishing returns). This also leads to different equilibria and potentially multiple equilibria, a situation in which coordination issues would be of primary interest. The role of networks and leaders would take a much different shape then than in the analysis provided here.

4. The values are computed in a way that amounts to adding up "apples and oranges" : p_i is the sum of risks from being close to other, potentially infectious, individuals, and of economic and social costs to avoiding activities. It makes no sense to add them, and could only correspond to a highly specific form of the individuals’ utility function, eliminating for instance risk aversion, decreasing returns, complementarities between health status and economic and social benefits, etc.

It cannot in any way represent risks that can vary with individual efforts, as these efforts have a different impact on the infection risk than on the economic impact. One cannot study an epidemic, which has a dynamic nature and is the source of specific anxiety to individuals, with parameters that sum infection risk and economic aspects.

In particular, the cost associated to a higher contact rate (the "alpha" part in the model) depends highly on the disease frequency in the population. The propagation of the epidemic follows a specific dynamic, endogeneous to individual activity choices, and the infection risk associated to a contact rate is not constant over time.

These assumptions are stated but not discussed in the paper, but they are crucial for the results.

The topic is of much importance given epidemiologists' concerns about future epidemics. I hope that some improvement can be done by building a different model. However the dynamics of the epidemic and the endogeneity of these dynamics to individual choices make this a highly difficult task. I unforuntately cannot suggest easy changes that would answer my concerns, but I do hope that you will find a way to build a different analysis with the same objective of analysing the impact of networls. I think it would be basically a different paper. If only one aspect is treated, at the very least the risks associated to a contact rate cannot be linear, cannot have a constant marginal value, in the face of an evolving epidemic, and cannot be added to economic benefits in a linear way. I hope that you will continue your efforts (that have been successful in the past) but with a better modeling of individual choices as a standard population game may not be suited here.

6. PLOS authors have the option to publish the peer review history of their article (what does this mean?). If published, this will include your full peer review and any attached files.

Reviewer #1: No

Reviewer #2: No

---

## [Author Response · Author response to Decision Letter 0]

11 Apr 2024

The response to the reviewers is uploaded as a pdf file in Response to Reviewers.

---

## [Decision Letter · Decision Letter 1]

23 Jul 2024

Beyond six feet:  The collective behavior of social distancing

PONE-D-23-33255R1

Dear Dr. Wu,

We’re pleased to inform you that your manuscript has been judged scientifically suitable for publication and will be formally accepted for publication once it meets all outstanding technical requirements.

Kind regards,

Vincent Antonio Traag, Ph.D.

Academic Editor

PLOS ONE

Additional Editor Comments (optional):

Reviewers' comments:

Reviewer's Responses to Questions

**Comments to the Author**

1. If the authors have adequately addressed your comments raised in a previous round of review and you feel that this manuscript is now acceptable for publication, you may indicate that here to bypass the “Comments to the Author” section, enter your conflict of interest statement in the “Confidential to Editor” section, and submit your "Accept" recommendation.

Reviewer #1: All comments have been addressed

2. Is the manuscript technically sound, and do the data support the conclusions?

Reviewer #1: Yes

3. Has the statistical analysis been performed appropriately and rigorously? 

Reviewer #1: Yes

4. Have the authors made all data underlying the findings in their manuscript fully available?

Reviewer #1: No

5. Is the manuscript presented in an intelligible fashion and written in standard English?

Reviewer #1: Yes

6. Review Comments to the Author

Reviewer #1: After evaluating the revised manuscript "Beyond six feet: The collective behavior of social distancing," including the authors' responses to my comments and concerns of the initial review process, I find that the manuscript has undergone significant improvements. Most notably, the Introduction is now clearer and better structured. It includes a more detailed discussion of relevant literature, providing a solid foundation for the study. The examples used are more accessible and relatable, enhancing the overall readability. The Results section has greatly benefitted from more elaborate descriptions of the model and the narrative style of presentation. The introduction of the logistic function is a valuable addition, providing a more realistic foundation for the model. The Discussion has been improved by contextualizing the study within existing behavioral theories. The inclusion of references to the theory of planned behavior and health belief models is sound and provides more context with regard to behavioral theories (one of my main concerns). However, the discussion of the results in terms of their real-world implications and comparisons with other studies remains somewhat superficial. While the study by Woike et al. has been referenced, a deeper comparison with the current study's findings would have strengthened this section. Despite some remaining concerns about the depth of the discussion regarding the implications and comparisons of the results, the manuscript has undergone significant improvements. I therefore recommend the revised manuscript for publication in PLOS ONE.

7. PLOS authors have the option to publish the peer review history of their article (what does this mean?). If published, this will include your full peer review and any attached files.

Reviewer #1: **Yes: **Hendrik Nunner

---

## [Editor Report · Acceptance letter]

26 Jul 2024

PONE-D-23-33255R1 

PLOS ONE

Dear Dr. Wu, 

I'm pleased to inform you that your manuscript has been deemed suitable for publication in PLOS ONE. Congratulations! Your manuscript is now being handed over to our production team.

Kind regards, 

on behalf of

Dr. Vincent Antonio Traag 

Academic Editor

PLOS ONE